# Diagnostic Efficacy of Advanced Ultrasonography Imaging Techniques in Infants with Biliary Atresia (BA): A Systematic Review and Meta-Analysis

**DOI:** 10.3390/children9111676

**Published:** 2022-10-31

**Authors:** Simon Takadiyi Gunda, Nonhlanhla Chambara, Xiangyan Fiona Chen, Marco Yiu Chung Pang, Michael Tin-cheung Ying

**Affiliations:** 1Department of Health Technology and Informatics, The Hong Kong Polytechnic University, Hung Hom, Kowloon, Hong Kong SAR 999077, China; 2Department of Rehabilitation Sciences, The Hong Kong Polytechnic University, Hung Hom, Kowloon, Hong Kong SAR 999077, China

**Keywords:** biliary atresia, ultrasonography, diagnostic accuracy, intraoperative cholangiography (IOC), diagnostic performance, elastography

## Abstract

The early diagnosis of biliary atresia (BA) in cholestatic infants is critical to the success of the treatment. Intraoperative cholangiography (IOC), an invasive imaging technique, is the current strategy for the diagnosis of BA. Ultrasonography has advanced over recent years and emerging techniques such as shear wave elastography (SWE) have the potential to improve BA diagnosis. This review sought to evaluate the diagnostic efficacy of advanced ultrasonography techniques in the diagnosis of BA. Six databases (CINAHL, Medline, PubMed, Google Scholar, Web of Science (core collection), and Embase) were searched for studies assessing the diagnostic performance of advanced ultrasonography techniques in differentiating BA from non-BA causes of infantile cholestasis. The meta-analysis was performed using Meta-DiSc 1.4 and Comprehensive Meta-analysis v3 software. Quality Assessment of Diagnostic Accuracy Studies tool version 2 (QUADAS-2) assessed the risk of bias. Fifteen studies consisting of 2185 patients (BA = 1105; non-BA = 1080) met the inclusion criteria. SWE was the only advanced ultrasonography technique reported and had a good pooled diagnostic performance (sensitivity = 83%; specificity = 77%; AUC = 0.896). Liver stiffness indicators were significantly higher in BA compared to non-BA patients (*p* < 0.000). SWE could be a useful tool in differentiating BA from non-BA causes of infantile cholestasis. Future studies to assess the utility of other advanced ultrasonography techniques are recommended.

## 1. Introduction

Biliary atresia (BA) is a congenital, inflammatory, destructive cholangiopathy affecting infancy and is characterised by progressive fibrosis and obliteration of both the intrahepatic and extrahepatic bile ducts [1,2]. The continued obliteration of the bile ducts and failure to restore biliary drainage is reported to progress to cholestasis, hepatic fibrosis, cirrhosis, end-stage liver failure, and, eventually, death if no liver transplantation is performed [3,4]. Clinically, infants with BA present with cholestatic jaundice, pale stool and dark urine that goes beyond the neonatal period [5]. The worldwide incidence of BA is reported to vary across the geographic plain, ranging from 1.5 per 10,000 live births in Taiwan [6] to 1 per 18,400 live births in France [7], with high incidence in East Asian countries such as China and Japan [8]. Despite BA being an uncommon disease, it is associated with high morbidity and mortality if undiagnosed and if treatment is delayed.

The Kasai portoenterostomy (KPE) is the primary treatment option for biliary atresia [2], to which success of the KPE procedure minimises the need for liver transplantation up to adulthood [4]. The success of KPE is, however, age-dependent, with a 2-month age at KPE being observed to result in high biliary recanalization and stabilization success rates of (80%) [1] and jaundice disappearance [9,10]. To mitigate the detrimental effects of poor prognosis arising from delayed age at KPE and misdiagnosis that may prompt unwarranted KPE, there is a need for an early and accurate differential diagnosis of BA.

Biliary atresia is diagnosed using several methods at present, including, but not limited to, clinical history, liver biopsy, medical imaging techniques such as intraoperative cholangiography (IOC), endoscopic retrograde cholangiography (ERC), and ultrasonography [10]. According to Chen et al. [11], IOC is the current strategy for the accurate diagnosis of BA. However, it is invasive and less accessible and could lead to considerable morbidity. Ultrasonography is a non-invasive, non-ionising, reliable, readily available, and cost-effective imaging modality [12], which is used as a screening and diagnostic tool for paediatric cholestasis evaluation to exclude biliary atresia [13]. Although there are no ultrasounographic features that are definitive of the diagnosis of BA, the triangular cord sign, gallbladder length, gallbladder morphologic characteristics, absence of gallbladder, and the presence of hepatic subcapsular flow are among some of the traditional signs consistent with BA diagnosis at ultrasound [14,15]. Ultrasonography imaging techniques have advanced in recent years, with several emerging techniques such as elastography, three and four-dimension (3D/4D) ultrasound, contrast-enhanced ultrasound (CEUS), and artificial intelligence that enable improved structural, hemodynamic, and functional evaluations of various organs [16]. The clinical utility of these advanced ultrasonography techniques is reported in various conditions and patient groups. Three-dimensional ultrasound was demonstrated to be comparable to magnetic resonance urography in the assessment of renal parenchymal volume [17], whereas, in obstetrics [18], prostate, and breast imaging 3D ultrasound becomes part of routine practice in adult imaging with undebated diagnostic and patient management benefits [19]. The application of 3D ultrasound in neonatal ventricular volume assessment is also promising [20]. CEUS utilises intravascular microbubble agents to delineate perfusion abnormalities that are linked to different pathological conditions, such as tumors and brain ischemia, among others, for which conventional ultrasound is limited [21].

Despite the demonstration of the clinical performance of the recent advanced ultrasonography techniques in various subjects and conditions, their efficacy in the diagnosis of biliary atresia is, however, understudied. This systematic review and meta-analysis are, therefore, aimed at evaluating the efficacy of the recent technological advances in medical ultrasonography in the diagnosis of infantile biliary atresia. It is hypothesised that the current available evidence demonstrates the clinical utility of the recently developed ultrasonography imaging techniques for the diagnosis of infantile biliary atresia.

## 2. Materials and Methods

The study involved searching the following six electronic databases; Cumulative Index to Nursing and Allied Health Literature (CINAHL Complete via EbscoHost), Medline, PubMed, Google Scholar, Web of Science (core collection), and Embase. The Hong Kong Polytechnic University online library was used to access these databases, with the last search performed on the 11 October 2022. The study followed the Preferred Reporting Items for Systematic Review and Meta-analysis (PRISMA) 2020 guidelines, as informed by Page et al. [22].

### 2.1. Search Strategy

The search strategy adopted involved searching the databases for the four concepts derived from the PICO framework structured research question, where *p* (study population) = infants with biliary atresia; I (intervention) = ultrasonography imaging technique that includes advanced ultrasonography techniques such as shear wave elastography, artificial intelligence, and grayscale ultrasonography mode; C (comparison) = liver biopsy representing the gold standard by which the accuracy level was compared, or cholangiography; O (outcome) = indicators of diagnostic performance (diagnostic accuracy, sensitivity, and specificity). The concepts were searched using (1.) MeSH descriptors in Medline and Pubmed, Emtree terms in Embase and CINAHL subject headings, and (2.) keywords and their related terms (synonyms, hyponyms). The reference section of the selected articles was also searched for relevant studies. The boolean operator “OR” was used to search within each PICO element concept (MeSH or Emtree terms) and the related entry terms or synonyms, whereas “AND” was utilised to search the concepts of the three PICO framework elements (population, intervention, and outcome). The database search strings are shown in Appendix A.

### 2.2. Inclusion and Exclusion Criteria

The studies included in the systematic review and meta-analysis were (1.) peer-reviewed studies involving humans and published in the English language from inception to 11 October 2022, (2.) studies assessing the diagnostic accuracy of advanced ultrasonography techniques such as shear wave elastography (SWE) in the differential diagnosis of biliary atresia from other causes of infantile cholestasis, (3.) studies assessing the diagnostic accuracy of algorithms combining the advanced ultrasonographic features such as shear wave velocity (SWV) with traditional diagnostic features from grayscale and color Doppler ultrasound such as the triangular cord sign, (4.) studies in which the measures of diagnostic performance are represented by the following diagnostic performance measures: overall accuracy, sensitivity, specificity, likelihood ratio, positive predictive values, negative predictive value and the area under the receiver operating curve (AUCROC) [23], (5.) studies in which parental consent and institutional ethical approval were obtained prior to data collection.

Exclusion criteria are studies with (1.) no gold standard used to compare diagnostic accuracy such as liver biopsy and surgical confirmation, (2.) inadequate information on study-population characteristics such as age and gender, (3.) inadequate information on the diagnostic performance outcome measures, (4.) conference proceedings, posters, case reports, reviews, editorial letters or commentaries, (5.) diagnostic accuracy measures of other imaging modalities and not ultrasonography, and (6.) non-English.

### 2.3. Data Extraction

Two reviewers, SG and NC, independently screened the title and abstract of the studies from the search strategy to exclude irrelevant articles and performed a full-text evaluation to check for eligible articles that were included in the systematic review and meta-analysis, whilst the third reviewer, MY, was responsible for resolving any disagreements. The data extraction form based on the PRISMA 2020 guidelines [22] was used to extract data on the authors, date of publication, study methodology (information for the assessment of the risk of bias, study settings, subject’s characteristics, ultrasonography features assessed, the gold standard against which the index test was compared), and the measures of diagnostic accuracy among other items from the eligible studies. The random effects model was used to determine the sensitivity and specificity of each ultrasound characteristic.

### 2.4. Quality Assessment

The Quality Assessment of Diagnostic Accuracy Studies tool version 2 (QUADAS-2) was utilised to assess the risk of bias and the methodological quality. The risk of bias and applicability concerns in the eligible studies were assessed on the following four domains, mainly, the (1.) selection of the patients, (2.) index test (3.) reference standard and (4.) flow and timing. The risk of bias and applicability in these domains were categorised into high, low or unclear, and meta-analysis was performed in studies that exhibited a low risk of bias in the assessed four domains.

### 2.5. Statistical Analysis

The Comprehensive Meta-Analysis Software version 3.3.070 was used for the comparison of the liver stiffness measurements between BA and non-BA patients through computing the effect size and confidence intervals for both individual studies and meta-analyses. These were displayed in the forest plots. Inconsistences in the studies were assessed using the I-square value and Chi-squared statistics (Q) tests, together with the qualitative assessment of the forest plots, whereas publication bias was assessed using the same software and depicted as a Funnel plot of inverse standard error by standardized difference in means. The effect size was expressed as the standardised mean difference (SMD) for the two continuous liver stiffness measures of (SWV and SWE kPa) between the BA and non-BA patients using a random effect model based on the mean, standard deviation and sample size. Hozo et al. Hozo, Djulbegovic [24]’s formula was used to convert the median and interquartile range (IQR) values to the respective means and standard deviations (SD) for studies in which the outcome measure was reported as median and IQR values. The Meta-DiSc (version 1.4, the Unit of Clinical Biostatistics team of the Ramón y Cajal Hospital in Madrid (Spain) was used to perform the pooling of the sensitivity, specificity and other diagnostic performance measures, utilising the incorporated DerSimonian-Laird random effect model.

## 3. Results

### 3.1. Literature Search

The initial database search strategy retrieved a total of 1060 records, as follows: Pubmed (n = 384), Embase (n = 247), CINAHL-Medline (n = 195), Web of Science (n = 34) and Google Scholar (n = 200) as shown in Figure 1. A total of 542 duplicates were identified and removed, and the remaining 518 records underwent a title and abstract screening. The title and abstract screening process excluded a total of 412 articles based on several reasons with the majority of the studies being excluded for utilising imaging modalities other than ultrasonography for the diagnosis of BA. These imaging modalities included magnetic resonance imaging, scintigraphy, computed tomography angiography, and endoscopic retrograde cholangiopancreatography [25,26,27]. Studies involved non-imaging diagnostic tests, such as the duodenal tube test [28,29], anti-smooth muscle antibodies and liver enzymes biomarkers [30,31] were also excluded. A total of thirteen systematic reviews and meta-analysis studies were excluded as they did not meet the publication criteria of original research articles, with some studies reporting the wrong population and index items. These studies consisted of five meta-analyses [32,33,34,35,36], and eight combined systematic reviews and meta-analyses [37,38,39,40,41,42,43,44]. Three of the meta-analysis focused on summarising the evidence on the diagnostic performance of various conventional ultrasound parameters for the diagnosis of biliary atresia [33,34,35]. In addition, Sun et al. [36] systematic review and meta-analysis was excluded, as it focused on evaluating the utility of hepatic subcapsular flow using conventional color doppler ultrasonography techniques, whereas, in the study of Guo et al. [32], meta-analysis was excluded as it assessed the diagnostic accuracy of acoustic radiation impulse force in the staging of hepatic fibrosis, not in the diagnosis of BA. Despite assessing the diagnostic performance of shear-wave elastography in differentiating BA from non-BA cases of jaundice, two studies [45,46] were excluded as the full-text articles were not available.

The remaining 106 studies underwent full-text article review and a total of 52 studies were excluded as they did not assess the diagnostic accuracy of advanced ultrasound imaging techniques but focused on the conventional grayscale ultrasound and Doppler techniques, where parameters such as gallbladder abnormalities, triangular cord sign, and the presence or absence of hepatic subcapsular flow on color Doppler ultrasound were assessed as predictor variables in the diagnosis of BA. A total of eight studies that were excluded during the full article review, which assessed the diagnostic performance of the advanced ultrasonography technique of SWE, not for the preoperative diagnosis of BA but the preoperative diagnosis of hepatic fibrosis in children with BA [47,48,49,50,51,52,53,54], whereas one study assessed the preoperative diagnostic accuracy of 2D SWE for liver cirrhosis in BA patients [55]. The other excluded studies focused on the diagnosis of liver fibrosis among post-operative BA children using elastography techniques [54], whereas two studies were excluded due to wrong outcomes as they evaluated the accuracy of the elastography techniques in the diagnosis of liver fibrosis, and this was for participants outside the specified age-range of the present study, which focused on infants below 1 year of age [56,57]. A study by Zhou et al. [58] that utilized an ensembled deep learning model, which was reported to surpass human expertise in the diagnosis of BA, was, however, excluded from the study, as the model was based on conventional sonographic gallbladder images. Two studies were excluded as they assessed the diagnostic utility of the contrast-enhanced ultrasound technique in percutaneous ultrasound-guided cholecysto-cholangiography among infants with BA [59,60].

Finally, a total of fifteen studies published in peer-reviewed journals, up to the 11 October 2022, met the eligibility criteria, in which the advanced ultrasonography technique of elastography was used for the diagnosis of BA among infants presenting with cholestasis (Figure 1).

### 3.2. Study Characteristics

The fifteen included studies were mainly single center, and consisted of twelve prospective cohort study designs [11,61,62,63,64,65,66,67,68,69,70,71], and four retrospective designs [11,72,73,74]. One study consisted of both prospective and retrospective study designs [11]. The total number of patients from the included studies was 2185, of which 50.6% of the patients were diagnosed with BA (n = 1105) whilst 49.4% were non-BA (n = 1080) patients. The main patient characteristics are presented in Table 1 and Figure 2.

Elastography ultrasound was the only advanced ultrasonography technique utilised in the included studies. The two indicators of liver stiffness measurements were shear wave velocity (SWV) (m/s) and the hepatic Young’s modulus (SWE kPa). The detailed individual study characteristics are shown in Table 2.

### 3.3. Diagnostic Performance of Shear Wave Elastography

The liver stiffness measurements (LSM) between BA and non-BA patients in the fifteen studies are shown in Table 3. A total of twenty-seven analyses were made and are depicted in Figure 3. Eight of the studies performed repeated analyses as they utilised different shear wave elastography modes, or categorised patients into different age groups. Among the twenty-seven analyses, fifteen were carried out in studies measuring the SWE kPa (Figure 4) whilst the remaining twelve analyses were performed for those studies in which the SWV was the outcome measure (Figure 5). The results from these analyses are presented below; first, for studies measuring the individual SWE parameters (SWE kPa and SWV) and, lastly, for all the studies to evaluate the diagnostic performance of the SWE ultrasonography technique.

#### 3.3.1. Diagnostic Performance of the Hepatic Young Modulus (SWE KPa)

Nine of the included studies assessed the diagnostic efficacy of the liver stiffness indicator, the hepatic Young’s modulus of elasticity (SWE kPa) in the assessment of cholestatic infants for the differential diagnosis of BA [62,63,64,68,69,70,71,72,73]. A total of four studies reported the liver stiffness measurements as mean SWE kPa [63,68,70,73], whilst five studies reported the SWE kPa median values [62,64,69,71,72] (Table 3). The difference in the median SWE kPa values between the BA and non-BA groups was statistically significant in all the studies utilizing the median SWE kPa values, with higher values observed in the BA group. Statistically significant findings (*p* < 0.01) were also observed between the mean SWE kPa values of BA and non-BA patients [63,68,73]. The cut-off point for the diagnosis of BA differed between the studies that used the mean SWE kPa values, and the reported cut-off values of the three studies were 12.35, 8.68, and 9.5 kPa, respectively [63,68,73]. The hepatic SWE kPa performance in the differential diagnosis of BA from cholestatic hepatic disease was reported to outperform that of conventional ultrasound parameters [63,75] with AUC values of 0.89 (95% CI: 0.829–0.935, *p* < 0.001 versus 0.748 (95% CI: 0.670–0.815, *p* < 0.001) in Wang et al. [75]. Contrary to these findings, one study observed that the diagnostic accuracy of SWE kPa does not surpass that of conventional grayscale ultrasound (AUC = 0.790 versus 0.893, respectively) [69]. The differentiating ability of SWE liver stiffness is reported to increase with the patient’s age (days) at diagnosis [72,73]. Liu et al. [72], reported an AUC of 0.91 in the (30–45) versus an AUC of 0.74 in the (15–30), whereas Shen et al. [73] noted an AUC of 0.905 in the (91–120) versus AUC of 0.761 in the less than 60 days age group. Similarly, Zhou et al. [69] observed that the diagnostic performance of SWE kPa in patients of less than 60 days of age (AUC = 0.694, 95% CI: 0.579–0.793) was lower than that in patients of greater than 60 days of age (AUC = 0.779, 95% CI: 0.682–0.858). However, contrary to the above findings, only one study by Boo et al. [62] observed a lower diagnostic accuracy of 80% in older patients compared to the diagnostic accuracies of 92.9%, 95.2%, and 100% in the younger age groups (<30, 31–60 and 61–90) days, respectively. The results from the present meta-analysis showed statistically significant differences in liver stiffness SWE kPa values, with higher values observed in BA patients compared to the non-BA cholestatic patients.

The pooled statistics showed an overall effect size, indicated by the SMD, of 3.018 kPa with, 95% CI of 2.256–3.779 (*p* < 0.0001) (Figure 4). Shear wave elastography kPa demonstrated good discriminatory abilities between BA and non-BA patients. The observed pooled diagnostic performance was: sensitivity = 0.83 (95% CI: 0.80–0.86); specificity = 0.80 (95% CI: 0.77–0.82); AUC = 0.9066; DOR = 26.92 (95% CI: 13.34–54.34) (Figure 6a,b,c and d, respectively).

#### 3.3.2. Diagnostic Performance of Shear Wave Velocity (SWV in m/s)

Six studies demonstrated the clinical utility of shear wave velocity (SWV) in discriminating BA from other causes of infantile cholestasis [11,61,65,66,67,74]. Four of these studies [11,61,65,67] reported the sensitivity, specificity, and diagnostic accuracy (AUC) of the SWV elastography modes (VTQ and VTIQ), whereas two studies [66,74] reported only the *p*-values indicating statistically significant differences between the mean SWV of BA and non-BA patients, with no information on the other diagnostic performance measures. Contrary to the findings of Liu et al. [61], in which a higher diagnostic accuracy value of 0.918 (95% CI: 0.834–1), was observed in the mean SWV of the VTIQ modes (cut-off SWV = 1.92 m/s). Sandberg et al. [67] reported a moderate diagnostic accuracy value of 0.7 (95% CI: 0.7–0.8) at a median cut-off SWV of 2.0 m/s. A three-color risk stratification model was developed in which five variables, including an SWV greater than 1.35 m/s, had a high accuracy in discriminating BA infants from non-BA infants (AUC= 0.983, sensitivity = 98.7% and specificity = 91.4%) [11]. Regardless of whether the mean SWV [11,61,66,74] or median SWV values [65,67] was used, all six studies reported that the SWV of the liver in the BA group was significantly higher than that in the non-BA (Table 3).

Biliary atresia patients exhibited significantly higher liver stiffness values as indicated by the SWV compared to non-BA patients (*p* < 0.0001). The pooled effect size (SMD) and 95% confidence intervals were 1.99 and (95% CI:1.487 to 2.494), respectively (Figure 5). The studies however showed high inconsistencies, I^2^ = 94.378. The pooled diagnostic performance for these studies was: sensitivity = 0.82 (95% CI: 0.80–0.84); specificity = 0.75 (95% CI: 0.72–0.78); AUC = 0.71; DOR = 18.22 (95% CI: 7.78–45.04) as shown in Figure 7a,b,c and d, respectively.

#### 3.3.3. Diagnostic Performance of the Combined Studies

The statistical analysis of the combined studies showed that BA patients had higher liver stiffness values compared to non-BA patients. Shear-wave elastography showed high discriminative power in the diagnosis of BA. The overall SMD and 95% confidence intervals of all the studies evaluating the diagnostic performance of BA were 2.516 and (2.084 to 2.947), respectively, *p* < 0.001. The studies showed considerable heterogeneity, I^2^ value of 95.079% (Figure 3). The pooled diagnostic performance was as follows: sensitivity = 0.83 (95% CI: 0.81–0.84); specificity = 0.77 (95% CI: 0.75–0.79); AUC = 0.896; DOR = 22.87 (95% CI: 13.16–39.75) as shown in Figure 8a, b, c and d, respectively.

The pooled standardised mean difference of the six studies that evaluated the differentiating ability of SWV was 1.990 (95% CI: 1.487–2.494); however, considerable heterogeneity was observed in these studies, I^2^ value of 94.378%, and, hence, a random effect model was adopted. A total of 11 out of the 12 analyses showed statistically significant differences between BA and non-BA patients’ liver stiffness measures (SWV), whereas only one analysis from Sandberg et al. [67], utilising the L9 probe and the VTIQ mode, had an insignificant result (*p* = 0.160) and was seen to reach the line of no effect (Figure 3 and Figure 5). The extracted data on the diagnostic performance indicators for the evaluated studies is shown in Table 4.

### 3.4. Studies Methodological Quality Assessment by the QUADAS Tool

The methodological quality of the studies was generally high, with the majority of studies exhibiting a low patient selection bias, as the consecutive selection of the participants was conducted in all studies except three [61,65,74] in which non-consecutive patient selection was used and unclear in each study, respectively (Table 5). In all the eligible studies, a case–control design was avoided. The inclusion–exclusion criteria were clear in all the studies, with low concerns of the selected patients not matching the review question that is focused on the diagnostic efficacy of advanced ultrasonography techniques for the diagnosis of biliary atresia among cholestatic infants. The reference tests were undertaken by different teams, who were blinded to the index test in all the studies; however, different reference standards were used, which could be a source of inhomogeneities. Despite the absence of an inter/intra-observer variability analysis among the two physicians who undertook the liver stiffness measurements for the index test in one study [73], the risk of bias in conducting the index test was deemed low, as the two physicians were reported to have more than five years of experience in abdominal ultrasonography. The reference standard was, however, not specified in one study [65]. The applicability concerns in the three domains of patient selection, index test, and reference standard were low in the majority of studies, except for two studies [69,70], in which three different reference standards (surgical exploration, intraoperative cholangiography under laparoscopy, or liver biopsy) were used to confirm the diagnosis of BA. The study flow timing in Liu et al. [72] was unclear; hence, it could have introduced bias as the time between the index test and reference test is not specified in the study.

The funnel plot in Figure 9, showed an asymmetrical distribution of the studies effects size, with the bottom of the plot showing a higher concentration of small studies only on one side of the mean effect size, demonstrating the small-study effects phenomena.

### 3.5. Publication Bias Assessment

The possibility of publication bias was assessed using the funnel plot shown in Figure 9.

## 4. Discussion

The early and accurate diagnosis of BA, to rule out other causes of infantile cholestasis, is important for better prognostic outcomes. The current strategy for the differential diagnosis of BA from non-BA causes of infantile cholestasis involves invasive procedures such as intraoperative cholangiography [11]. The need for non-invasive accurate diagnostic tests, therefore, cannot be overemphasised. Ultrasonography is a non-invasive imaging technique and has seen several advances in its technology over recent years that have the potential to improve the differentiation of BA from non-BA causes of cholestasis in infants [16]. Systematic reviews of the diagnostic performance of conventional grayscale ultrasound techniques have been reported [36,42]. To the best of our knowledge, this is the first study to summarise the available evidence on the diagnostic performance of advanced ultrasonography techniques in the differential diagnosis of BA from other causes of infantile cholestasis.

The study results showed that only one advanced ultrasound imaging technique, shear wave elastography, was studied, to assess its diagnostic performance for the preoperative diagnosis of BA (Table 2). There are no studies assessing the diagnostic efficacy of other recent ultrasonography advances, such as microvascular imaging and contrast-enhanced ultrasound, that met the inclusion criteria. The two studies related to microvascular imaging technique in this review, however, were excluded from the analysis after a full article review, as they evaluated the clinical utility of MVI in a post-KPE procedure in BA patients and not for the preoperative diagnosis of BA [77]. The ability of MVI to detect capsular flows that conventional color Doppler could not among the BA group in the study by Lee et al. [77] is an indicator of its possible diagnostic utility among preoperative BA patients. It is, therefore, prudent to have studies assessing the diagnostic accuracy of MVI for the preoperative diagnosis of BA.

The significantly higher liver stiffness values observed in BA patients in comparison to non-BA patients; (overall SMD, 95% confidence intervals and *p* values) of (2.578, (2.136–3.02) and *p* < 0.0001), respectively (Figure 3), is an indicator that the shear wave elastography-based liver stiffness measurement can facilitate the differentiation of BA from other causes of infantile cholestasis. The technique involving an L9 probe in the VTIQ mode was assessed in only one study [67], and exhibited poor discriminatory ability (*p* = 0.16); hence, more studies are required to evaluate the clinical utility of this technique in discriminating BA from non BA cholestatic infants before concluding its relevance for clinical use. The effect size was higher in studies in which SWE (kPa) was the outcome measure, with an overall SMD of 3.08 in SWE kPa studies versus 2.078 for SWV studies.

The current study observed a good diagnostic performance of SWE with the pooled sensitivity of 0.83 (95% CI: 0.81–0.84), specificity of 0.77 (95% CI: 0.75–0.79), AUC of 0.896, and DOR of 22.87 (95% CI: 13.16–39.75) (Figure 8a–d). These findings are in agreement with those from a recent meta-analysis by Wagner et al. [40], which evaluated the diagnostic performance of SWE in which high diagnostic accuracy was reported (AUC = 0.91) versus the current study’s AUC of 0.896. The results from the meta-analysis demonstrated that ultrasound-based liver stiffness assessment could be a valuable imaging marker for the diagnosis of infantile biliary atresia. It is, however, imperative to note that, despite the current study reporting SWE to have good diagnostic accuracy, the diagnostic performance of SWE does not exceed that of conventional grayscale ultrasound parameters, as reported from pooled studies in a meta-analysis by Yoon et al. [42], where the overall diagnostic accuracy (AUC = 0.97) for conventional grayscale parameters was higher than that for SWE reported in the current study (AUC = 0.896). The results from the systematic review showed that combining SWE and grayscale ultrasound yields a better diagnostic specificity [63], and similar findings were echoed by Wang et al. [71], who concluded that, despite the hepatic Young’s modulus being an independent predictor of BA, the incorporation of the gallbladder structural features and age into a nomogram realized a better performance than the individual features. The subgroup analysis observed notable excellent diagnostic accuracy in studies utilising the hepatic Young’s modulus compared to those reporting shear wave velocity: AUC was 0.906 versus 0.71, respectively. The systematic review demonstrated that the diagnostic performance of SWE increased with age and this can pose a potential clinical challenge in the utility of SWE for the early diagnosis of BA in young patients, which is key to obtaining good prognostic outcomes, as suggested by Napolitano [1].

It is imperative to note that the methodological approaches in the included studies were varied, as different machines, scanning protocols, reference index, and outcome measures were utilized (Table 2). Six of the studies used the Aixplorer ultrasound system (SuperSonic Imagine SA, Aix-en-Provence, France) [69,70,72,73,75], five studies used the Acuson S2000 or S3000 unit (Siemens Healthcare, Erlangen, Germany) [11,61,65,66,67], and two studies used the TUS-Aplio 500 scanner (Toshiba Medical Systems, Tokyo, Japan) [63,70]. The FibroScan 502 Touch (Echosens, Paris, France), in conjunction with a 5 MHz probe, was used in two studies [62,64]. Moreover, the measurement outcomes were reported differently in the studies, with some studies reporting mean values and others reporting median values (Table 3). These differences could account for the heterogeneities observed in the current study I^2^ = 87.2%, chi square = 156.46%, *p* < 0.0001 (Figure 8a). In one of the studies, different diagnostic performances were reported across different SWE modes (VTIQ and VTQ), probes, and scanning regions of interest (ROI) [67]. The liver stiffness measures were also not uniformly reported, as these were reported either as mean or median values. Hence, to facilitate the meta-analysis, the median values were converted to the mean values using established formula [24]. The current study findings point toward the need for future standardization of SWE protocols for the diagnosis of BA, which will allow for an accurate pooling of the studies of diagnostic performance.

The bias assessment is represented by the funnel plot in Figure 9. The concentration of low-precision studies shown at the base of the plot is an indicator that there are more small studies reported in comparison to large-sample-size studies. The observed funnel plot asymmetry is indicative of the small-study effect phenomena, in which all of the evaluated low-precision studies are observed to concentrate only on the positive side of the mean effect size. These findings could suggest the presence of publication bias, although they do not rightly imply that publication bias was present [78,79,80] as funnel plot asymmetry may be due to other causes, including but not limited to, between-study heterogeneity and chance [80].

## 5. Limitations of the Study

The study, however, is limited, as only a few studies with a small number of patients met the inclusion criteria, which could restrict the generalizability of the study results. It should be noted that the included studies were mainly limited to the Asian and American population, further limiting the external validity of the results to other populations. The possibility of publication bias is another limitation of this study, as this could lead to an overestimated observed mean effect size. The evaluated studies utilised different reference standards and there were inconsistencies in outcome reporting, with some studies reporting mean values, whereas others reported median values, which could be a source of inhomogeneities.

## 6. Conclusions

The results from the current systematic review and meta-analysis have demonstrated that shear wave elastography has a good diagnostic performance and could, therefore, be a useful complementary tool to other diagnostic methods in differentiating BA from non-BA causes of infantile cholestasis. Liver stiffness indicators were significantly higher in BA patients compared to non-BA patients. Future studies assessing the utility of other advanced ultrasonography techniques are recommended.

## Figures and Tables

**Figure 1 children-09-01676-f001:**
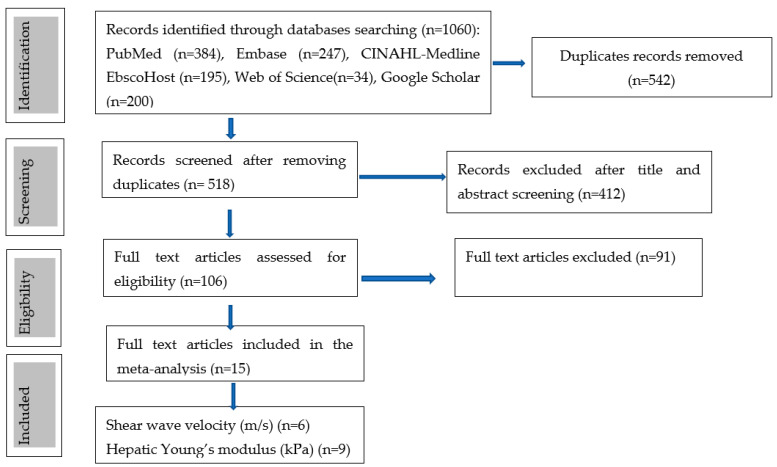
Study selection process (flow chart diagram).

**Figure 2 children-09-01676-f002:**
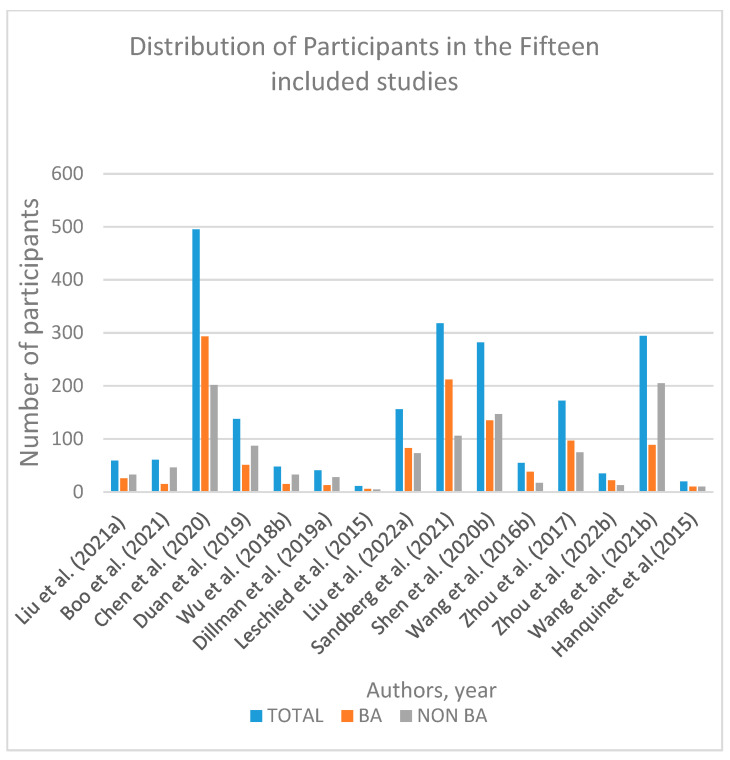
Frequency distribution of participants in the fifteen included studies [11,61,62,63,64,65,66,67,68,69,70,71,72,73,74].

**Figure 3 children-09-01676-f003:**
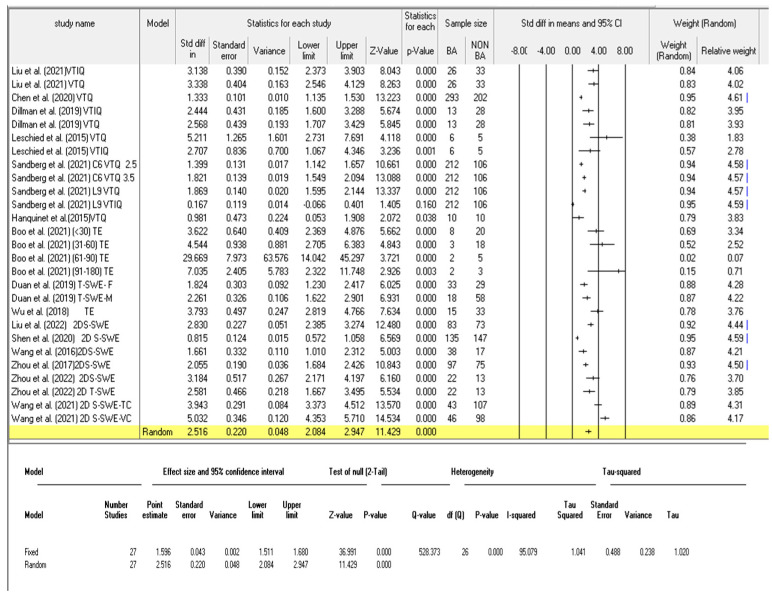
A comparison of the liver stiffness value between the patients with and without BA for all the studies (KPa and SWV) [11,61,62,63,64,65,66,67,68,69,70,71,72,73,74].

**Figure 4 children-09-01676-f004:**
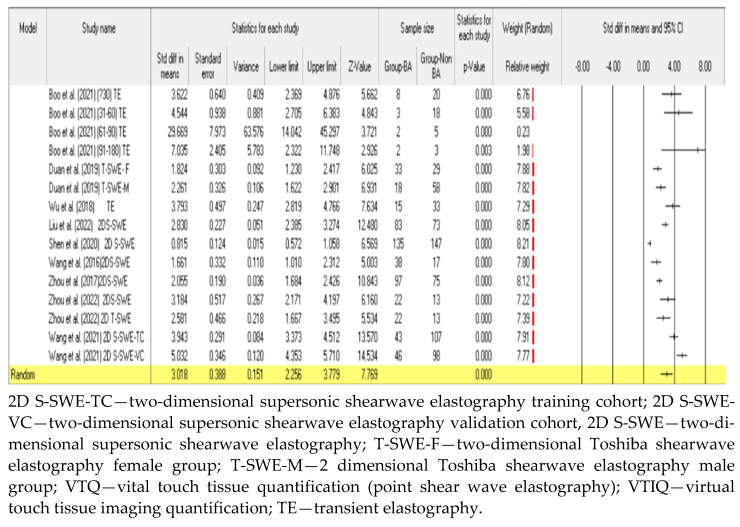
A comparison of the liver stiffness parameter (kPa) between BA and non-BA patients [62,63,67,68,69,70,71,72].

**Figure 5 children-09-01676-f005:**
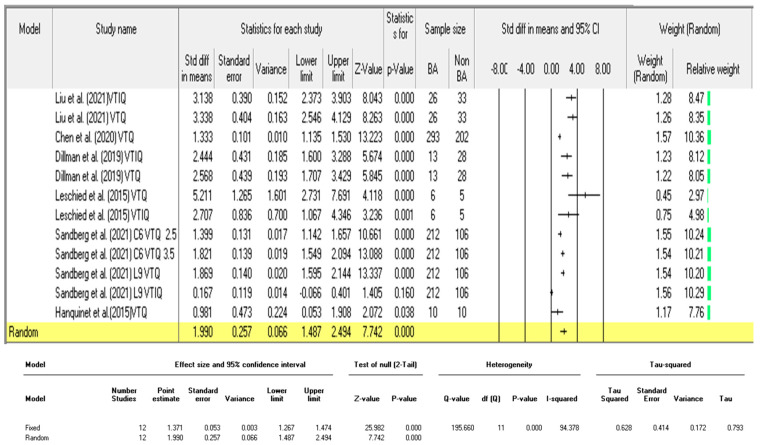
A comparison of the liver stiffness parameter (SWV) between BA and non-BA patients [11,61,65,66,67,74].

**Figure 6 children-09-01676-f006:**
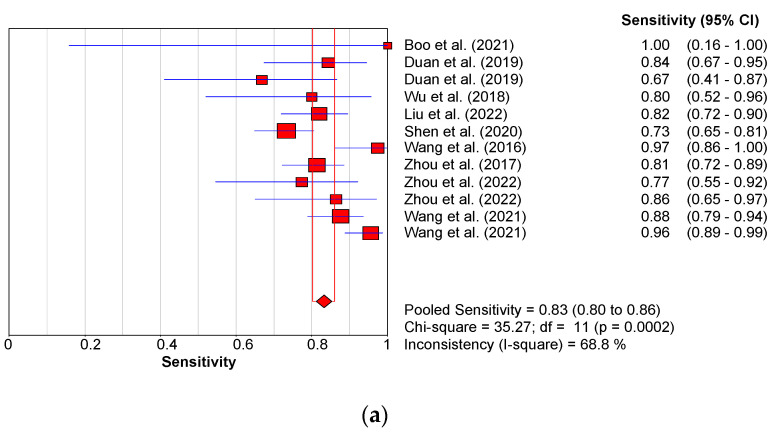
(**a**): Sensitivity forest plot for studies using SWE (kPa). (**b**): Specificity forest plot for studies using SWE (kPa). (**c**): Summary receiver operating characteristic curve for studies using SWE (kPa). (**d**): Forest plot showing overall diagnostic odds ratio for studies using SWE (kPa) [62,63,64,68,69,70,71,72,73].

**Figure 7 children-09-01676-f007:**
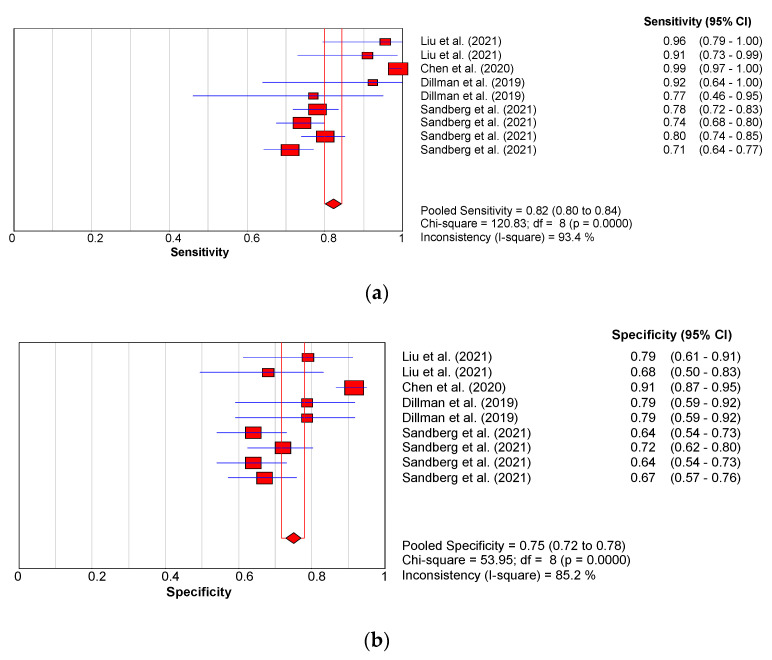
(**a**): Sensitivity forest plot for studies using shear wave velocity. (**b**): Specificity forest plot for studies using shear wave velocity. (**c**): Summary receiver-operating characteristic curve for studies using shear wave velocity. (**d**): Forest plot showing overall diagnostic odds ratio for studies using shear wave velocity. [11,61,65,67].

**Figure 8 children-09-01676-f008:**
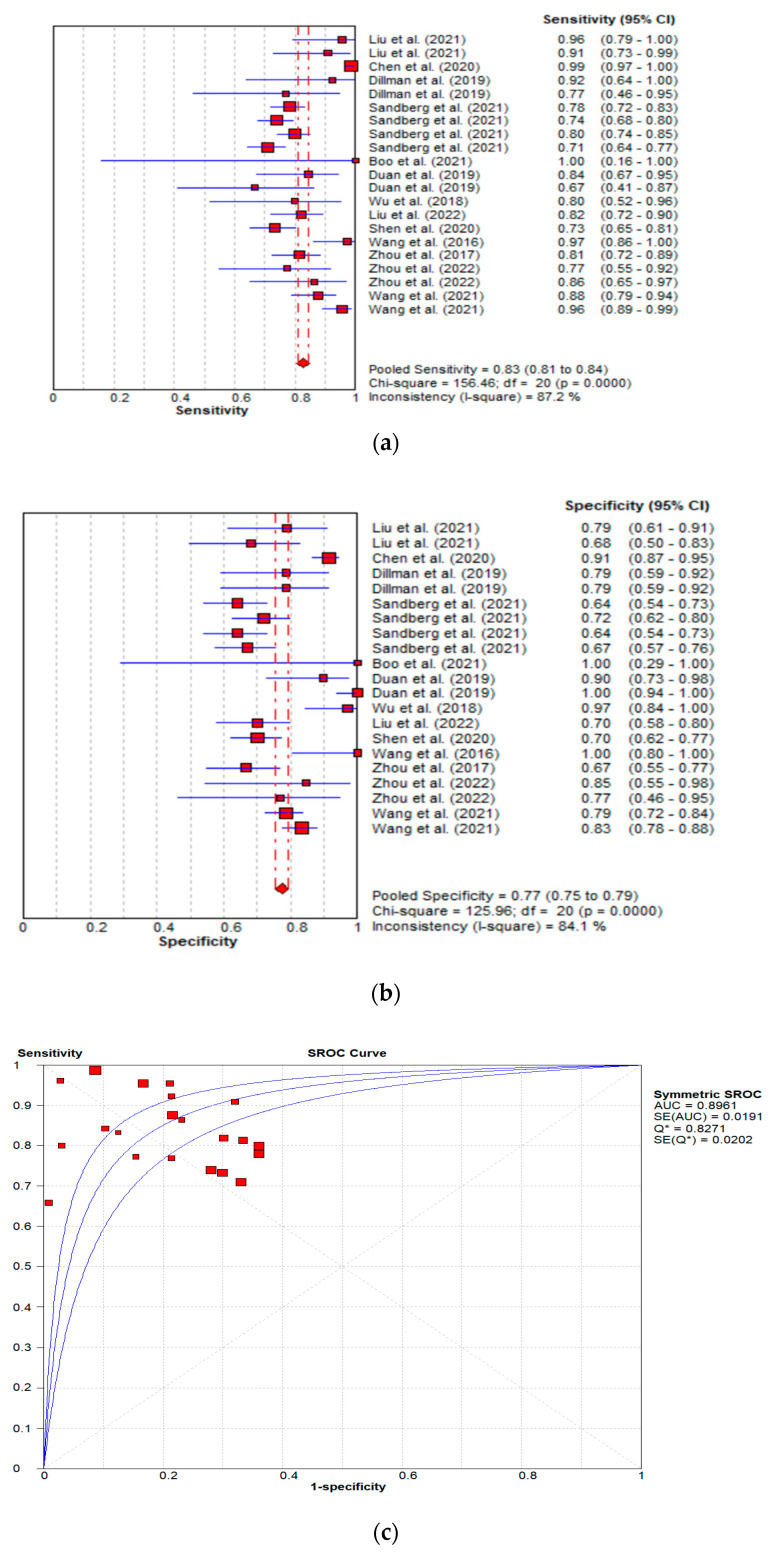
(**a**): Sensitivity forest plots for all included studies. (**b**): Specificity forest plots for all included studies. (**c**): Summary receiver operating characteristic curve for all included studies. (**d**): Forest plot showing overall diagnostic odds ratio for all included studies [11,61,62,63,64,65,67,68,69,70,71,72,73].

**Figure 9 children-09-01676-f009:**
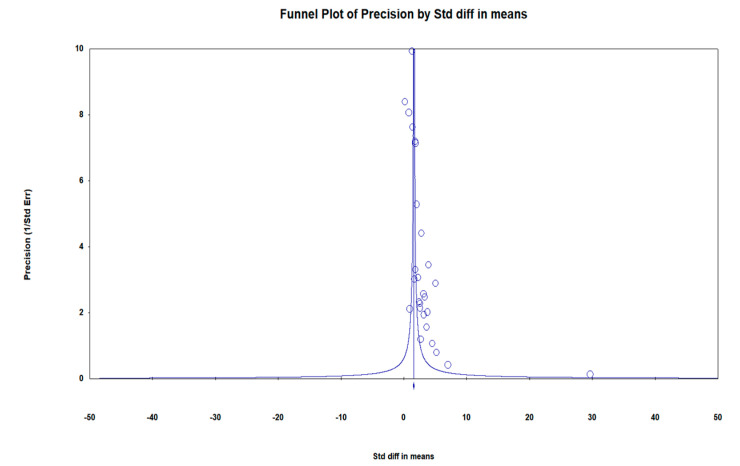
Funnel plot of precision (inverse standard error) by standardised difference in means assessing bias.

**Table 1 children-09-01676-t001:** Main patient characteristics of the included studies.

Author(s), Year	Ref	Country	Type of Patient	Patients (n)			Age (Days)		
				Total (n = 2185)	BA (n = 1105)	Non-BA (n = 1080)	Overall Age	BA	Non-BA
Liu et al. (2021)	[61]	China	Infants with Cholestasis	59	26	33	NA	^a^ 72.5 ± 29.0 (30–127)	^a^ 81.3 ± 35.2 (25–141)
Boo et al. (2021)	[62]	Taiwan	Cholestatic infants	61	15	46	NA	^a^ 45 (13–121); ^b^ 30 (22–63)	^b^ 35.5 (24–51.3)
Chen et al. (2020)	[11]	China	Cholestatic infants	495 (^R^ 308; ^P^ 187)	293 (^R^ 186; ^P^ 107)	202 (^R^ 122; ^P^ 80)	^a^ 52.4 ± 19.5	^aR^ 55.2 (19.7) ^aP^ 51.0 (18.8)	^aR^ 51.9 (18.8) ^aP^ 45.8 (26.8)
Duan et al. (2019)	[63]	China	Cholestatic hepatitis	138	51	87	NA	^c^ 43 (5–88) days	^c^ 30 (5–90)
Wu et al. (2018)	[64]	Taiwan	Cholestatic infants	48	15	33	^a^ 45.87 (9–87)	^b^ 45 (34.5–60.5)	^b^ 40 (27–56)
Dillman et al. (2019)	[65]	USA	Neonatal cholestasis	41	13	28	^b^ 37 (24–52)		
Leschied et al. (2015)	[66]	USA	Infantile liver disease	11	6	5	^a^ 107 (42–336)	^a^ 79 (range 42–196)	^a^ 140 (56–336)
Liu et al. (2022)	[72]	China	Infantile cholestasis	156	83	73	^b^ 36 days (25–41)	NA	NA
Sandberg et al. (2021)	[67]	China	Cholestatic jaundice	318	212	106	NA	^a^ 59.7 ± 18.8 (20–114)	^a^ 65.7 ± 25.6 (9–186)
Shen et al. (2020)	[73]	China	Cholestatic jaundice	282	135	147	NA	^a^ 59 ± 18.8	^a^ 70 ± 20.4
Wang et al. (2016)	[68]	China	Cholestatic hepatitis	55	38	17	NA (16–140)	^a^ 42	^a^ 50
Zhou et al. (2017)	[69]	China	Cholestatic infants	172	97	75	NA	^a^ 65.3 ± 20.5 (26–134)	^a^ 62.4 ± 22.0 (2–140)
Zhou et al. (2022)	[70]	China	Cholestatic infants	35	22	13	NA	^b^ 61 (45–75)	^b^ 69 (50–87)
Wang et al. (2021)	[75]	China	Cholestatic infants	294 (^T^ 150; ^V^ 144)	89 (^T^ 150; ^V^ 144)	205 (^T^ 150; ^V^ 144)	^a^ 42.94 (4–67)	^bT^ 46 (33–54) ^bV^ 50 (33–57)	^bT^ 47 (33–54) ^bV^ 44 (33–57)
Hanquinet et al. (2015)	[74]	Switzerland	Cholestatic infants	20	10	10	52.1 ± 29.2	NA	NA

BA—Biliary atresia; Non-BA—No Biliary atresia; NA—not available; ^a^ mean Age ± standard deviations, with ranges in parentheses; ^b^ median age, with interquartile range (IQR) in parentheses; ^c^ median age with range in parentheses; ^R^ Patients from retrospective study; ^p^ Patients from prospective study; ^aR^ mean age in the retrospective group; ^aP^ mean age in the prospective group; ^T^ Patients from the training cohort; ^V^ Patients from the validation cohort; ^bT^ median age training group; ^bV^ median age validation group.

**Table 2 children-09-01676-t002:** Main study characteristics of the included studies.

Author(s), Year.	Ref	Study Design	Type of Ultrasoumd Machine	Ultrasound Technique	Reference Standard	Ultrasound Parameter
Liu et al. (2021)	[61]	Prospective single center cohort	Siemens Acuson OXANA2 (Siemens Healthcare, Erlangen, Germany) with a 3–5.5 MHz 6C1 convex transducer probe and a 4–9 MHz 9L4 linear array probe.	VTQ and VTIQ	Surgical exploration	mean VTQ & VTIQ SWV
Boo et al. (2021)	[62]	prospective cohort study	TE (FibroScan 502 Touch; Echosens, Paris, France), S1 probe (5 MHz)	TE	IOC, surgical	Median TE kPa
Chen et al. (2020)	[11]	Prospective and retrospective analysis single center	Siemens Acuson S2000 (Siemens Medical Solutions) with a 4- to 9-MHz linear transducer.	VTQ	IOC and Intraoperative biopsy	median VTQ SWV
Duan et al. (2019)	[63]	Prospective, single center.	TUS-Aplio 500 scanner (Toshiba Medical Systems, Tokyo, Japan).14L5 linear array probe (10 MHz)	VTIQ	KPE and liver biopsy	Mean VTIQ kPa
Wu et al. (2018)	[64]	Prospective, single center	TE (Fibroscan 502 Touch; Echosens, Paris, France) S1 probe (5 MHz)	TE	IOC and liver biopsies	Median (kPa)
Dillman et al. (2019)	[65]	Prospective, multi-center study.	Acuson S2000 or S3000 (Siemens Healthcare, Erlangen, Germany); 9L4 linear transducer probe	VTIQ and VTQ	Not specified	Median VTQ and VTIQ SWV
Leschied et al. (2015)	[66]	Single-center retrospective	Acuson S3000 US system/ 9L4 transducer (Siemens Medical Solutions USA, Malvern, PA)	VTQ and VTIQ	liver biopsy andIOC	1. mean VTQ and VTIQ
Liu et al. (2022)	[72]	single-center retrospective study	Aixplorer ultrasound system (SuperSonic Imagine SA, Aix-en-Provence, France) with an L15-4 linear probe	VTIQ	IOC and Biopsy	
Sandberg et al. (2021)	[67]	prospective cohort	(Siemens), with C6 and L9 transducers,	VTQ & VTIQ 2 transducers & 2 ROI	biopsy	Median SWV
Shen et al. (2020)	[73]	retrospective	Aixplorer ultrasound system(SuperSonic Imagine SA, Aix-en-Provence, France), and L15–4 linear probe.	VTIQ	Kasai surgery	mean SWE kPa
Wang et al. (2016)	[68]	Single-center case control	Aixplorer ultrasound system (SuperSonic Imagine SA, Aix-en-Provence, France), an L15-4 linear probe.	VTIQ	Kasai surgery	mean SWE kPa
Zhou et al. (2017)	[69]	Single-center prospective cohort study	AixPlorer scanner (Supersonic Imagine, Paris, France) with a(1 to 6 MHz curvilinear transducer and 4 to 15 MHz linear array transducer 2.A linear array transducer (SL15-4)	VTIQ	surgical exploration, IOC and liver biopsy	Median kPA
Zhou et al. (2022)	[70]	Single-center prospective cohort study	Aixplorer scanner (SuperSonic Imagine, Aix-en-Provence, France), linear array transducer SL15-4 (5 to 14 MHz). Toshiba T-SWE used Aplio500 (Canon Medical System, Otawara, Tochigi, Japan), a linear array transducer 14-L5 (5 to 14 MHz)	S-SWE and T-SWE	surgical exploration, IOC and liver biopsy	mean SWE kPa
Wang et al. (2021)	[75]	Single-center prospective analysis	1. Aixplorer US system (SuperSonic Imagine, Aix-en-Provence, France), with linear probe. 2. HI VISION Ascendus (Hitachi Medical Systems, Japan) equipped with a 5–13 MHz linear-array transducer	VTIQ (2D SWE) training and validation groups	IOC	Mean SWEkPa
Hanquinet et al. (2015)	[74]	retrospective	Acuson^®^ S2000 or S3000 US machine (Siemens Healthcare, Erlangen, Germany) a linear 9-MHz probe	VTQ	IOC & Liver biopsy	mean VTQ SWV

ROI—region of interest; IOC—Intraoperative cholangiography; VTQ—vital touch tissue quantification (point shear wave elastography); VTIQ—virtual touch tissue imaging quantification (2D-SWE); TE—transient elastography.

**Table 3 children-09-01676-t003:** Liver stiffness measurements (LSM) values among patients with and without BA.

Author(s), Year	Ref	Elastography Technique	Hepatic Young’s Modulus (kPA)		SWV(m/s)		Main Finding
			BA	Non BA	BA	Non BA	
Liu et al. (2021)	[61]	VTIQ	NA	NA	^a^ 2.43 ± 0.29	^a^ 1.52 ± 0.29	VTQ and VTIQ can help distinguish BA from non-BA; VTIQ has higher sensitivity and specificity than VTQ
	[61]	VTQ	NA	NA	^a^ 2.36 ± 0.36)	^a^ 1.30 ± 0.28	
Boo et al. (2021)	[62]	(≤30) TE	^b^ 8.4 (6.8–16.8) ^ab^ 10.1 (2.9)	^b^ 4.2 (3.3–5.4) ^ab^ 4.3 (0.64)			Statistically significant difference between BA and non-BA TE values. A cutoff LSM > 7.7 kPa had high diagnostic accuracy for BA in all age groups, except for the group of 91–180 days of age.
	[62]	(31–60) TE	^b^ 10 (5.5–13) ^ab^ 9.63 (2.15)	^b^ 5.4 (3.8–6.2) ^ab^ 5.2 (0.72)			
	[62]	(61–90) TE	^b^ 19.4 (19.1–19.7) ^ab^ 19.4 (0.27)	^b^ 5.5 (4.5–6.1) ^ab^ 5.4 (0.51)			
	[62]	(91–180) TE	^b^ 40.8 (26–55.5) ^ab^ 40.78 (8.52)	^b^ 3.8 (3.2–8.8) ^ab^ 4.9 (1.65)			
Chen et al. (2020)	[11]	VTQ			^a^ 1.77 (0.39)	^a^ 1.30 (0.29)	Mean SWV is significantly higher in BA than in other causes of cholestasis, *p*< 0.001.
Duan et al. (2019)	[63]	T-SWE-VTQ	^a^ 17.59 ± 5.65	^a^ 9.84 ± 1.49			Both SWE and grayscale ultrasound have good performance in diagnosing BA. SWE increases the diagnostic specificity when combined with grayscale ultrasound.
	[63]	T-SWE VTIQ)	^a^ 17.94 ± 6.44	^a^ 9.91 ± 2.00			
Wu et al. (2018)	[64]	TE	^b^ 10.50 (8.50–20.90) ^ab^ 12.6 (3.61)	^b^ 4.60 (3.90–6.00) ^ab^ 4.78 (0.64)			LSM assessment during the workup of cholestatic infants may facilitate the diagnosis of BA.
Dillman et al. (2019)	[65]	2DSWE VTIQ			^b^ 2.08 (1.90–2.50) ^ab^ 2.14 (0.27)	^b^ 1.49 (1.34–1.80) ^ab^ 1.53 (0.24)	SWV were significantly different between BA and non-BA subjects, *p* = 0.0001 SWE showed better diagnostic performance for distinguishing BA from non-BA causes of neonatal cholestasis, *p* = 0.0014.
	[65]	Point SWE VTQ			^b^ 1.95 (1.48–2.42) ^ab^ 1.95 (0.34)	^b^ 1.21 (1.12–1.51) ^ab^ 1.26 (0.23)	
Leschied et al. (2015)	[66]	VTQ			^a^ 2.08 ± 0.17 (1.90–2.30)	^a^ 1.28 ± 0.13 (1.09–1.44)	A significant difference between the VTQ mean SWV of the BA and non-BA groups, *p*< 0.0001. The mean color pixel values were significantly different between BA and non-BA subjects, *p* < 0.0001.
	[66]	VTIQ			^a^ 3.14 ± 0.73 (2.24–4.40)	^a^ 1.61 ± 0.23 (1.34–1.87)	
Liu et al. (2022)	[72]	S-SWE	^b^ 9.37 (7.30–11.45) ^ab^ 9.37 (1.22)	^b^ 6.50 (5.95–7.65) ^ab^ 6.65 (0.53)			LSM measurement by SWE & Serum GGT level showed the best performances for differentiating BA fromNon BA LSM diagnostic value increased with age, and AUC = 0.91 in patients of (30–45) versus 0.74 in 1 (5–30) days old, *p* < 0.01.
Sandberg et al. (2021)	[67]	C6 VTQ 2.5			^b^ 1.9 (1.6–2.3) ^ab^ 1.93 (0.29)	^b^ 1.59 (1.3–1.7) ^ab^ 1.55 (0.23)	SWE had significantly better performance in differentiating BA from non-BA cases when compared to grayscale ultrasound.
	[67]	C6 VTQ 3.5			^b^ 1.9 (1.6–2.3) ^ab^ 1.93 (0.29)	^b^ 1.4 (1.3–1.7) ^ab^ 1.45 (0.2)	
	[67]	L9 VTQ			^b^ 2.1 (1.7–2.4) ^ab^ 2.08 (0.29)	^b^ 1.5 (1.3–1.9) ^ab^ 1.55 (0.27)	
	[67]	L9 VTIQ			^b^ 2.2 (1.9–2.5) ^ab^ 2.2 (2.7)	^b^ 1.8 (1.6–2.1) ^ab^ 1.83 (0.25)	
Shen et al. (2020)	[73]	2D S-SWE	^a^ 12 (6.0)	^a^ 8.1 (3.3)			Parallel testing of GGT and LSM in infants < 90 days decreases the rate of BA misdiagnosis, *p* < 0.001.
Wang et al. (2016)	[68]	2D S-SWE	^a^ 20.46 ± 10.19	^a^ 6.29 ± 0.99			Mean SWE values were significantly higher in BA than non-BA hepatitis syndrome and control groups, *p*< 0.01.
Zhou et al. (2017)	[66]	2D S-SWE	^b^ 12.6 (10.6–18.8) ^ab^ 13.65 (2.39)	^b^ 9.6 (7.5–11.7) ^ab^ 9.6 (1.23)			Diagnostic performance of LSM values in identifying BA was lower than that of grayscale ultrasound, *p* < 0.001. S-SWE was comparable to T-SWE (AUC 0.895 vs. 0.822, *p* = 0.071) in diagnosing BA. T-SWE had good performances in the diagnosis of BA and the assessment of liver fibrosis compared with S-SWE, *p* < 0.002.
Zhou et al. (2022)	[70]	2D S-SWE	^a^ 14.0 (11.1–20.0)	^a^ 8.2 (7.1–9.7)			
	[70]	2D T-SWE	^a^ 11.0 (9.1–13.5)	^a^ 8.5 (6.5–9.2)			
Wang et al. (2021)	[75]	2D S-SWE (TC)	^b^ 9.9 (8.4–14.3) ^ab^ 10.63 (1.7)	^b^ 6.6 (5.7–7.5) ^ab^ 6.6 (0.56)			Age (*p* = 0.009), gallbladder morphology (*p* = 0.001) and hepatic elasticity (*p* < 0.001) are independent predictive factors to differentiate between BA and other causes of cholestasis.
	[75]	2D S-SWE (VC)	^b^ 11.1 (8.7–12.8) ^ab^ 10.93 (1.2)	^b^ 6.3 (5.3–7.7) ^ab^ 6.4 (0.72)			
Hanquinet et al. (2015)	[74]	VTQ			^a^ 2.2 (0.4)	^a^ 1.7 (0.6)	Significance difference between BA and non-BA SWV, *p* = 0.049

^a^ mean values (SD); ^b^ median values (IQR); ^ab^ calculated mean from median values using formulae by Hozo et al. [24]; SWV—shear wave velocity; KPA—hepatic Young’s modulus; 2D S-SWE—2 dimensional supersonic shearwave elastography; 2D T-SWE—2 dimensional Toshiba shearwave elastography; VTQ—vital touch tissue quantification (point shear wave elastography); VTIQ—virtual touch tissue imaging quantification (2D-SWE); TE—transient elastography; LSM—liver stiffness measurement; TC—Training cohort; VC—Validation cohort.

**Table 4 children-09-01676-t004:** Diagnostic performance indicators for each study.

Author (s)	Ref	Elastography Technique	Cutoff Value	Sen (%)	Spec (%)	PPV (%)	NPV (%)	AUC	DA	BA(n)	Non-BA(n)	TP	TN	FP	FN
Liu et al. (2021)	[61]	VTIQ	1.92	95.5	78.9	NA	NA	0.92	NA	26	33	24.83	26.04	6.96	1.17
	[61]	VTQ	1.77	90.9	68	NA	NA	0.89	NA	26	33	23.63	22.44	10.56	2.37
Chen et al. (2020)	[11]	VTQ	1.35	98.7	91.4	94	98.1	0.98	93.6	293	202	289.1	184.63	17.37	3.809
Dillman et al. (2019)	[65]	2DSWE VTIQ	1.84	92.3	78.6	66.7	95.7	0.89	NA	13	28	12	22.01	6	1.00
	[65]	Point SWE VTQ	1.53	76.9	78.6	62.5	88	0.81	NA	13	28	9.99	22.01	5.99	3.003
Leschied et al. (2015)	[66]	Mean SWV VTQ	NA	NA	NA	NA	NA	NA	NA	6	5	NA	NA	NA	NA
	[66]	Mean SWV VTIQ	NA	NA	NA	NA	NA	NA	NA	6	5	NA	NA	NA	NA
Sandberg et al. (2021)	[67]	C6 VTQ 2.5	1.5	78	64	NA	NA	0.8	NA	212	106	165.36	67.84	38.16	46.64
	[67]	C6 VTQ 3.5	1.6	74	72	NA	NA	0.8	NA	212	106	156.88	76.32	29.68	55.12
	[67]	L9 VTQ	1.6	80	64	NA	NA	0.8	NA	212	106	169.6	67.84	38.16	42.4
	[67]	L9 VTIQ	2	71	67	NA	NA	0.7	NA	212	106	150.52	71.02	34.98	61.48
Boo et al. (2021)	[62]	(91–180) TE	8.8	100	100	100	100	100	100	2	3	2	3	0	0
	[62]	(≤30) TE	7.7	NA	NA	100	90.9	NA	92.9	8	20	NA	NA	NA	NA
	[62]	(31–60) TE	7.7	NA	NA	100	94.7	NA	95.3	3	18	NA	NA	NA	NA
	[62]	(61–90) TE	7.7	NA	NA	100	100	NA	100	2	5	NA	NA	NA	NA
	[62]	(91–180) TE	7.7	NA	NA	66.7	100	NA	80	2	3	NA	NA	NA	NA
Duan et al. (2019)	[63]	T-SWE-VTQ-f	12.35	84.3	89.7	82.7	90.7	0.937	87.7	33	29	27.82	26.013	2.987	5.181
	[63]	T-SWE VTIQ-M	12.35	66.7	100	100	83.6	0.833	87.7	18	58	12.0	58	0	5.99
Wu et al. (2018)	[64]	TE	7.7	80	97	NA	NA	85.3	NA	15	33	12	32.01	0.99	3
Liu et al. (2022)	[72]	S-SWE	7.1	81.3	69.86	NA	NA	0.82	NA	83	73	68.00	50.9978	22.02	14.9981
Shen et al. (2020)	[73]	2D S-SWE	9.5	73.3	70.1	69.2	74.1	0.771	NA	135	147	98.96	103.047	43.95	36.045
Wang et al. (2016)	[68]	2DS-SWE	9.5	97.4	100	100	96.9	0.997	NA	38	17	37.01	17	0	0.988
Zhou et al. (2017)	[69]	2DS-SWE	10.2	81.4	66.7	76	73.5	0.79	NA	97	75	78.96	50.025	24.975	18.042
Zhou et al. (2022)	[70]	2DS-SWE	10.2	77.3	84.6	89.5	68.8	0.895	80	22	13	17.0	10.998	2.002	4.994
	[70]	2D T-SWE	8.7	86.4	76.9	86.4	76.9	0.822	82.9	22	13	19.0	9.997	3.003	2.992
Wang et al. (2021)	[75]	2D S-SWE-TV	7.81	87.6	78.5	63.9	93.6	0.888	81.3	89	205	77.964	160.925	44.075	11.036
	[75]	2D S-SWE-N	7.81	95.5	83.4	71.4	97.7	0.94	87.1	89	205	85	171	34.	4.00
Hanquinet et al. (2015)	[74]	VTQ	NA	NA	NA	NA	NA	NA	NA	10	10	NA	NA	NA	NA

NA—not available; Sen—sensitivity; Spec—specificity; PPV—positive predictive value; NPV—negative predictive value; DA—diagnostic accuracy AUC—area under receiver-operating curve; CI—confidence interval, 95% CI in parenthesis; mean values (TV—training + validation; N—normogram; TP—true positive; TN—true negative; FP—false positive; FN—false negative) were calculated based on the given data on sensitivity, specificity and the number of patients in each of the two groups (BA and non-BA groups) using Baratloo et al [76] formulae.

**Table 5 children-09-01676-t005:** QUADAS tool studies methodological quality assessment results (risk of bias and applicability concerns).

Aurthor(S), Year	Ref	Risk Of Bias				Applicability Concerns		
		Patient Selection	Index Test	Reference Standard	Flow and Timing	Patient Selection	Index Test	Reference Standard
Liu et al. (2021)	[61]	Unclear	Low	Low	Low	Low	Low	Low
Boo et al. (2021)	[62]	Low	Low	Low	Low	Low	Low	Low
Chen et al. (2020)	[11]	Low	Low	Low	Low	Low	Low	Low
Duan et al. (2019)	[63]	Low	Low	Low	Low	Low	Low	Low
Wu et al. (2018)	[64]	Low	Low	Low	Low	Low	Low	Low
Dillman et al. (2019)	[65]	High	Low	Unclear	Low	Low	Low	Unclear
Leschied et al. (2015)	[66]	Low	Low	Low	Low	Low	Low	Low
Liu et al. (2022)	[72]	Low	Low	Low	Unclear	Low	Low	Low
Sandberg et al. (2021)	[67]	Low	Low	Low	Low	Low	Low	Low
Shen et al. (2020)	[73]	Low	Low	Low	Low	Low	Low	Low
Wang et al. (2016)	[68]	Low	Low	Low	Low	Low	Low	Low
Zhou et al. (2017)	[69]	Low	Low	Low	Low	Low	Low	High
Zhou et al. (2022)	[70]	Low	Low	Low	Low	Low	Low	High
Wang et al. (2021)	[75]	Low	Low	Low	Low	Low	Low	Low
Hanquinet et al. (2015)	[74]	Unclear	Low	Low	Low	Low	Low	Low

Low = Low Risk; High = High Risk; Unclear = Unclear Risk.

## Data Availability

Not applicable.

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
