# Peer review of "Diagnostic Efficacy of Advanced Ultrasonography Imaging Techniques in Infants with Biliary Atresia (BA): A Systematic Review and Meta-Analysis"

_children, 2022, doi:10.3390/children9111676_

Round 1

Reviewer 1 Report

The authors submit a qualitatively excellent meta-analysis on the diagnostic value of ultrasonography in BA patients. The design of the study is exemplary and the conclusion formally correct. However, differential diagnosis in neonatal cholestasis must cover many different parameters simultaneously, and only in synopsis with the individual clinical course can the tentative diagnosis be addressed. In addition, we must refer to the hypothesis that BA is an ongoing inflammatory process, which reduces the relevance of punctual measurements with any technique. On the other hand, BA is defined as irreversible fibrotic obstruction of the extrahepatic bile ducts, which has to be confirmed by endoscopic, MIS- guided, or open cholangiography. From this point of view, it would not be advisable to refer just to one single diagnostic tool regardless of its statistical relevance. In other words, elastography is one diagnostic tool among many others but it is still far from making more invasive procedures obsolete. This has already been demonstrated in most of the series evaluated by the authors. Unfortunately, the meta-analysis of these papers doesn’t provide any new information and could, therefore cannot be recommended for publication.

Reviewer 2 Report

At first, I would like to congratulate the authors on their work. The authors have performed a systematic review on the utility of advanced Ultrasound techniques for the diagnosis of Biliary atresia.

The study has merit and will be of interest to our readers. However, I have several comments that need to be addressed.

Introduction: What did you hypothesize before conducting this research? Please write your hypothesis in 2-3 lines at the end of this section.

Methods:

-The search strategy is not optimal. Please see the link below and revise it:

https://systematicreviewsjournal.biomedcentral.com/articles/10.1186/s13643-017-0644-y

-The authors must provide the search strings used for each search engines (in a supplementary file)

-In section 2.4, the authors have mentioned- "The meta-analysis was performed in studies that did not exhibit significant publication bias whilst measuring similar diagnostic outcome of interest". What do you mean by it? The authors must perform a publication bias assessment and depict it in a funnel plot. 

Results: Well-written. I would advise the authors to expand the legends of the figures. The legends must be 2-3 lines and must depict the main findings of the figures.

Discussion: The authors must mention all limitations. Please comment about inconsistency, publication bias, etc.

Round 2

Reviewer 1 Report

The reviewer thanks the authors for their response. However, their arguments are not strong enough to change my mind. I´m sorry that I stay with my general recommendation and that this paper is from my view not relevant enough to be published in JCM

Reviewer 2 Report

In the revised manuscript, the authors have incorporated all my suggestions. The overall scientific quality of the manuscript has improved significantly. I would like to congratulate the authors for their work.